# How European Research Projects Can Support Vaccination Strategies: The Case of the ORCHESTRA Project for SARS-CoV-2

**DOI:** 10.3390/vaccines11081361

**Published:** 2023-08-14

**Authors:** Anna Maria Azzini, Lorenzo Maria Canziani, Ruth Joanna Davis, Massimo Mirandola, Michael Hoelscher, Laurence Meyer, Cédric Laouénan, Maddalena Giannella, Jesús Rodríguez-Baño, Paolo Boffetta, Dana Mates, Surbhi Malhotra-Kumar, Gabriella Scipione, Caroline Stellmach, Eugenia Rinaldi, Jan Hasenauer, Evelina Tacconelli

**Affiliations:** 1Infectious Diseases Division, Department of Diagnostics and Public Health, University of Verona, 37134 Verona, Italy; lorenzomaria.canziani@univr.it (L.M.C.); massimo.mirandola@univr.it (M.M.); evelina.tacconelli@univr.it (E.T.); 2Division of Infectious Diseases and Tropical Medicine, Medical Center of the University of Munich (LMU), 80802 Munich, Germany; hoelscher@lrz.uni-muenchen.de; 3German Center for Infection Research (DZIF), Partner Site Munich, 80802 Munich, Germany; 4Centre de Recherche en Epidemiologie et Sante des Population, Institut National de la Sante et de la Recherche Medicale (INSERM), Universite Paris-Saclay, 94270 Le Kremlin-Bicêtre, France; laurence.meyer@inserm.fr; 5INSERM, IAME UMR 1137, Universite Paris Cite, 75018 Paris, France; cedric.laouenan@inserm.fr; 6Departement d’Epidemiologie Biostatistiques e Recherche Clinique, AP-HP, Hospital Bichat, 75018 Paris, France; 7Infectious Diseases Unit, IRCCS Azienda Ospedaliero-Universitaria di Bologna, 40138 Bologna, Italy; maddalena.giannella@unibo.it; 8Department of Medical and Surgical Sciences, University of Bologna, 40138 Bologna, Italy; paolo.boffetta@unibo.it; 9Infectious Diseases and Microbiology Unit, Hospital Universitario Virgen Macarena and Department of Medicine, Biomedicines Institute of Sevilla-CSIC, University of Sevilla, 41004 Sevilla, Spain; jesusrb@us.es; 10Centro de Investigacion Biomedica en Red en Enfermedades Infecciosas, Instituto de Salud Carlos III, 28029 Madrid, Spain; 11Stony Brook Cancer Center, Stony Brook University, Stony Brook, NY 10041, USA; 12National Institute of Public Health, 050463 Bucharest, Romania; dana.mates@insp.gov.ro; 13Laboratory of Medical Microbiology, Vaccine & Infectious Disease Institute, University of Antwerp, 2000 Antwerp, Belgium; surbhi.malhotra@uantwerpen.be; 14Supercomputing Applications and Innovation Department, Cineca Consorzio Interuniversitario, 40033 Bologna, Italy; g.scipione@cineca.it; 15Berlin Institute of Health at Charite, Universitätsmedizin Berlin, 10117 Berlin, Germany; caroline.stellmach@bih-charite.de (C.S.); eugenia.rinaldi@bih-charite.de (E.R.); 16Life and Medical Sciences Institute, University of Bonn, 53113 Bonn, Germany; jan.hasenauer@uni-bonn.de; 17Institute of Computational Biology, Helmholtz Center Munich—German Research Center for Environmental Health, 85764 Neuherberg, Germany

**Keywords:** COVID-19, cohort study, public health, vaccination, data harmonization, fragile population, healthcare workers

## Abstract

ORCHESTRA (“Connecting European Cohorts to Increase Common and Effective Response To SARS-CoV-2 Pandemic”) is an EU-funded project which aims to help rapidly advance the knowledge related to the prevention of the SARS-CoV-2 infection and the management of COVID-19 and its long-term sequelae. Here, we describe the early results of this project, focusing on the strengths of multiple, international, historical and prospective cohort studies and highlighting those results which are of potential relevance for vaccination strategies, such as the necessity of a vaccine booster dose after a primary vaccination course in hematologic cancer patients and in solid organ transplant recipients to elicit a higher antibody titer, and the protective effect of vaccination on severe COVID-19 clinical manifestation and on the emergence of post-COVID-19 conditions. Valuable data regarding epidemiological variations, risk factors of SARS-CoV-2 infection and its sequelae, and vaccination efficacy in different subpopulations can support further defining public health vaccination policies.

## 1. Introduction

The SARS-CoV-2 pandemic has been the most severe threat to global health and economic prosperity since World War II. The development and implementation of effective prevention and therapeutic strategies are crucial components of the fight against SARS-CoV-2, alongside the development of vaccination strategies. Studies regarding real-world data on COVID-19 vaccine uptake and effectiveness in different sub-populations at higher risk of infection and mortality are essential. Optimizing the use of serology as a correlate of protection for vaccines [1] is of primary importance in developing public health strategies [2] and achieving group immunity.

Since the onset of the pandemic, the European Union has invested significant resources in EU-wide collaborative research projects with a view to a better understanding of the SARS-CoV-2 infection, of the efficacy of vaccines in the context of the emerging viral variants and as a means to support decision making at all levels, from global and public health down to individual patient care and patient–doctor shared decision making in terms of diagnostic and therapeutic algorithms and vaccination policy [3,4].

One such project is ORCHESTRA (“Connecting European Cohorts to Increase Common and Effective Response To SARS-CoV-2 Pandemic”, https://orchestra-cohort.eu/, accessed on 28 June 2023), whose main objective is to help rapidly advance the knowledge related to the prevention of the SARS-CoV-2 infection and the management of COVID-19 and its long-term sequelae [5]. It was funded under the EU research and innovation funding program Horizon 2020 and launched in December 2020. Cohort studies can quickly answer key scientific questions in the management of infectious diseases. ORCHESTRA, in particular, can provide scientific evidence useful for guiding policymakers in the context of public health decisions due to the rigorous methodology and the sample size of participating cohorts.

## 2. Materials and Methods

ORCHESTRA is a 3-year project which foresees the creation of a pan-European longitudinal cohort of patients with and without SARS-CoV-2 infection. It consists of four main studies (Figure 1), each composed of different cohorts from European and non-European countries, adhering to the ORCHESTRA consortium (see Appendix A).

The first study (“COVID-19 cohorts and long-term sequelae”) is focused exclusively on COVID-19 patients to assess the short and long-term outcomes of acute infection, also after specific therapeutic measures and/or prophylactic approaches.

The other three studies focus, respectively, on the general population, healthcare workers (HCWs) and fragile patients (namely, pregnant women, children, people living with HIV infection, solid organ transplant recipients, people affected by solid cancer or hematological malignancies, patients with immunologic disorders, and patients with Parkinson’s disease), and enroll both infected and uninfected subjects. Each of these studies investigates aspects of the SARS-CoV-2 infection specific to the population analyzed, such as incidence of infection and identification of specific risk factors.

Moreover, 4 patient registries, comprising aggregated data from more than 2 million people, joined the project for additional studies regarding population-level analysis.

The studies also assess the effect of the SARS-CoV-2 vaccination. Among the prospectively collected variables, there are the type and timing of administered vaccines, post-vaccination side effects, humoral and cell-mediated response to vaccination at pre-established time points, and breakthrough infections (BI), defined as a SARS-CoV-2 infection occurring despite the protection elicited by the vaccination. The impact of different viral variants on vaccine effectiveness has also been evaluated.

The main key methodological strengths of the project are the harmonized data collection across different cohorts and countries and the technical infrastructure of high-performance computing (HPC) facilities for processing data.

The key to creating a pan-European longitudinal cohort is the focus on harmonization and standardization of data [6]. Standard terminologies were used and updated to create a consistent semantic representation of over 2500 COVID-19-related variables. More than 700 variables were shared between different cohorts, so combined analysis is possible without further processing. New concept requests were submitted to standards development organizations, contributing to global interoperability. In the case of already-existing cohorts, variables were harmonized, while prospective cohorts adopted a standard and shared protocol.

To assess the epidemiological, clinical, microbiological, immunological, and genotypic aspects of the enrolled populations and environmental and socio-economic features, high computing performance and fast storage capabilities, provided by HPCs, were used to create a dedicated cloud infrastructure, fulfilling the requirements for data management confidentiality/privacy, integrity and security in compliance with the European GDPR regulations [7,8]. Three main layers compose the overall infrastructure: National Data Providers, National Hubs (one for each HPC center involved: CINECA—Italy, CINES—France and HLRS—Germany) to centralize data storage and processing, and the ORCHESTRA Data Portal for sharing aggregated data and results [5].

Ethics committee approval has been obtained for the master protocols and for each participating center. In addition, as a Horizon 2020-funded project, ORCHESTRA is subject to an ethics appraisal pby the European Commission. This includes an ethics review conducted prior to project commencement as well as ethics checks and audits during and after project implementation. The ethics review is conducted by independent experts appointed by the European Commission and focuses on compliance with ethical rules and standards, relevant European legislation, international conventions and declarations, national authorizations and ethics approvals, proportionality of the research methods, and awareness of the ethical aspects and social impact of the planned research. As a result of this review, ORCHESTRA has a contractual obligation to submit specific deliverables to the European Commission as part of the ethics checks and audits procedure. These include study protocols, informed consent templates, data protection impact assessment (DPIA) and details on the protection of vulnerable individuals, the anonymization/pseudonymization procedure of collected data, the data minimization principle, and on lawful basis for further data processing in case of secondary use of data.

## 3. Main Results

To date, nearly 60 historical and prospective cohorts from 18 countries worldwide have been recruited and established comprising more than 600,000 individuals (Figure 1) as well as up to 2 million individuals in patient registries (see Appendix A).

The ORCHESTRA prospective cohorts collect periodic samples for microbiological, virological, immunological, and epi-genomic analysis to supplement epidemiological and clinical data. The analysis of data from these cohorts has provided important insights into the characteristics of the COVID-19 pandemic with a special focus on the vaccine efficacy across different groups in terms of serological response and breakthrough infections.

### 3.1. SARS-CoV-2 Vaccination in HCWs

The two main benefits of HCW cohorts are the rapid enrollment of individuals with higher exposure to SARS-CoV-2, and early uptake of vaccines according to specific prevention regulations, enabling precise estimates of vaccine efficacy over time. The unified ORCHESTRA HCW cohort, including more than 64,000 HCWs from Italy, Spain, Slovakia, and Romania, described the impact of the SARS-CoV-2 vaccination on the determinants of the infection, supporting subsequent vaccination strategies in this setting [9]. For example, RNA-based vaccines and heterologous vaccination were associated with increased antibody levels, with a high level of confidence [9]. Moreover, this study design permits the description of uncommon outcomes in a vaccinated population, such as the occurrence of BI. A study from this cohort documented that the cumulative incidence of BI was 1.2%, and the risk was reduced in HCWs undergoing a complete vaccination cycle. The relationship between the antibody titer and the risk of BI is not direct: despite a waning of antibody titer after the second dose of vaccine, effective protection lasted at least six months after the second dose, regardless of the type of mRNA vaccine used. Moreover, after a full vaccination cycle, the BI manifested itself in a milder form [10].

In a subset of 4824 HCWs, anti-S antibody titer was described over time [11] and was dependent on previous COVID-19 infections. Interestingly, 900 HCWs (24%) experienced an Omicron infection in the period in which they received the third dose of an RNA-based vaccine and showed sustained immunity against subsequent infections compared to uninfected individuals (time ratio: 2.26, *p* < 0.001).

A complementary analysis of more than 7,700 HCWs showed that the surge of the Omicron variant caused an increase in the incidence of infections (2130; 60% of all registered infections) and re-infections (386; 95% of all registered infections) in already-vaccinated HCWs. Among the most important risk factors of infection, three doses of vaccine significantly reduced the disease onset (aHR 0.58 95%CI: 0.41; 0.80) [12]. Notably, working in COVID-19 areas does not seem a preponderant risk factor for infection (aHR 1.71; 95%CI 0.90; 3.25) due to the active protocols for personal protection and the higher perceived risks from the HCWs [12].

### 3.2. SARS-CoV-2 Vaccination in Fragile Populations

The ORCHESTRA fragile cohort has provided valuable information in support of tailored vaccination policies for specific high-risk populations [13,14,15].

In a study on serological response after a two-dose vaccine cycle, 52% of 1062 solid organ transplant recipients reached detectable antibody titers after 3 months [16]. Some immunosuppressive drugs such as mycophenolate were related to negative antibody response at 3 months on multivariable analysis (OR 0.29; 95%CI 0.20; 0.43), a result in line with other reports from different settings [17].

A subsequent analysis of more than 600 solid organ transplant recipients showed that even if 75% of patients reached immunization, about 20% of patients developed a BI, predominantly with mild to moderate clinical manifestations. Among the infected subjects, hospitalized patients were older, reported a higher rate of comorbidities, and had received the third dose of vaccine less frequently. Patients who developed BI were younger and had a shorter median time from transplantation, compared to those without a BI. Furthermore, patients with a heart transplant were estimated to have the lowest probability of reaching a positive antibody response and the highest risk of developing a BI. In terms of vaccine type, patients who received all three doses of mRNA-1273 showed a slightly higher probability of reaching immunization and the lowest probability of developing a BI [18,19]. 

Similarly, studies conducted within ORCHESTRA on people living with HIV (PLWH) have provided crucial scientific evidence to guide SARS-CoV-2 vaccination schemes within this population. Firstly, after two doses of vaccine, humoral immunogenicity is absent or poorly elicited in subjects with low CD4 count (CD4 < 200/mm^3^), strongly suggesting that this population would benefit from a third dose [20]. Similarly, the results show that a third dose should be considered also for PLWHs with a CD4 count between 200 and 500 cells/mm^3^ [20]. A second study conducted on 625 PLWHs who received a third vaccine dose showed that the additional dose elicited a strong humoral immune response and quickly compensated for the waning of antibody levels after the second dose. Furthermore, CD4 count was confirmed as a strong predictor of humoral response to SARS-CoV-2 vaccination in PLWH, giving a laboratory marker for targeted strategies for appropriate delivery of an eventual fourth dose in this population [21].

Moreover, the extensive systematic review undertaken by ORCHESTRA, regarding the impact of different underlying conditions of fragile patients on COVID-19 outcomes, provides valuable insight into the selection of priority populations for preventive interventions, such as vaccines and monoclonal antibodies [13,14,15].

### 3.3. SARS-CoV-2 Vaccination Campaign in the General Population

The ORCHESTRA general population cohorts focused on SARS-CoV-2 transmissibility and risk factors for infection in the context of the emergence of new viral variants and investigating epidemiological differences across different pandemic waves. An assessment of the circulation of the Omicron and Delta variants in France in December 2021 investigated the replacement time window between these two variants of concern, and the comparison of the different times of introduction of Omicron across regions [22]. The estimated rate R_0_ for Omicron to the R_0_ of Delta was between 1.57 and 2.34. The developed model enabled short-term analysis of the epidemiological characteristics of an emerging variant, and it can be easily used to describe more generally the evolution of epidemic diseases evolving into different variants. Such approaches provide knowledge that can support decision makers in evaluating risks caused by an epidemic and by emerging variants of concern of the same infective agent, carefully planning vaccination campaigns, as well as evaluating the future impact on public health systems, economies, and other affected areas.

The role of social determinants was explored in a national-level cohort in France to investigate how inequalities influence the risk of SARS-CoV-2 infection. In particular, seroprevalence rose from 4.5% in May 2020 to 6.2% in November 2020. In a context of less strict social restrictions, the rise was uneven and higher in the 18–24-year-old population (from 4.8% to 10.0%) and among second-generation immigrants from outside Europe (from 5.9% to 14.4%), which seems to have reinforced territorialized socialization among peers [23].

### 3.4. Impact of SARS-CoV-2 Vaccination on Post-COVID-19 Condition

The ORCHESTRA prospective cohort study of long-term sequelae in SARS-CoV-2-infected individuals is currently assessing drivers of post-COVID-19 condition (PCC). Specifically, after 1 year of follow-up, 57% of more than 1700 enrolled individuals comply with the WHO definition of PCC. The relative distribution of symptoms outlines four phenotypes with different risk factors and possibly representing different pathologic mechanisms: the chronic-fatigue-like, the respiratory, the chronic pain, and the neurosensorial phenotypes. Vaccination and early therapy with monoclonal antibodies during acute infection seem to have a substantial role in preventing chronic-fatigue-like PCC (under publication). In addition, the precise description of the clinical picture of PCC might improve its definition, as explored by other groups worldwide [24].

A comprehensive study of virus variants and vaccine types across multiple countries was conducted based on the integration of harmonized epidemiological data via computational models and made possible through the ORCHESTRA methodological approach [6]. The model assessed the optimization of vaccine allocation and non-pharmaceutical interventions. It predicted that a non-uniform vaccine allocation can be highly beneficial, for both risk groups and at the country level, regardless of the use of non-pharmaceutical interventions. This implies that policymakers can improve measures by applying strategic vaccine allocations, even when strong non-pharmacological interventions are applied [25].

## 4. Discussion

As a cohort study, ORCHESTRA naturally counteracts selection bias by looking at different subgroups of subjects from different countries. Furthermore, adopting a specific reference point, such as the emergence of breakthrough infection after SARS-CoV-2 vaccination, allows for the estimation of the absolute risk of infection with a new viral variant, since the observation is longitudinal over time. Each ORCHESTRA cohort can support the evaluation of risk for the population of interest, as well as the differences in risk between different groups, and provide data to determine potential risk factors. Moreover, a cohort study allows for the examination of multiple, even unpredictable, effects, and by collecting data during regular time intervals, it is possible to indicate more clearly the temporal sequence between exposure and outcome to minimize recall error. Finally, the typical limitations of a cohort study, such as inadequacy for issues with long latencies, the need for a large population to generate enough data points, and the need for extended follow-up periods for the outcome to manifest, are much less relevant in a pandemic context, strengthening the study design of the ORCHESTRA project.

With this premise, ORCHESTRA described the immunogenicity of the COVID-19 vaccination and its protective effect against severe disease, both in healthy [10,11] and fragile subjects, rapidly after the introduction of vaccines [18,19]. Other studies have reported results that are in line with ORCHESTRA. For example, in one cohort study conducted in Switzerland after the emergence of Omicron [26], individuals with natural immunity had 15 times lower antibody titers than vaccinated people, and less than half of them showed neutralization activity. In another cohort study enrolling HCWs [27], triple vaccination and double vaccination with prior or subsequent infection showed the highest neutralization response. Scientific evidence from all these studies can be of relevance both in informing public health vaccination strategies and in addressing vaccine hesitancy.

In conclusion, notwithstanding the intrinsic limitations of cohort studies, this type of study represents one of the most feasible designs to explore risk factors, genome associations, public health interventions, long-term sequelae, and vaccination effectiveness in a pandemic situation. Importantly, they can provide early information to design randomized clinical trials and play an essential role in data harmonization among different settings and countries. The ORCHESTRA project highlights how EU-funded research, which by design involves multiple countries and settings, heterogeneous data sources, and multidisciplinary teams, can provide access to multifaceted and extensive data. The Horizon 2020 funding program has been critical to sustain independent research which is essential for public support for public health strategies, particularly in the case of a very polarized public debate such as that surrounding SARS-CoV-2 vaccination.

## Figures and Tables

**Figure 1 vaccines-11-01361-f001:**
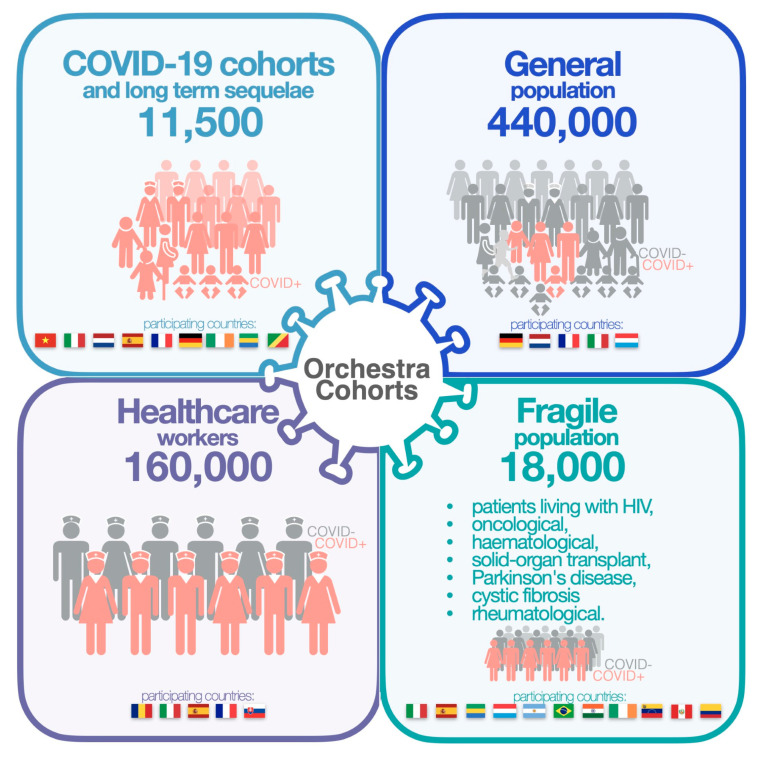
Overview of the ORCHESTRA project.

## Data Availability

Dataset partially available at this link https://orchestra-cohort.eu/public-data-set/, accessed on 28 June 2023.

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
