# Peer review of "How European Research Projects Can Support Vaccination Strategies: The Case of the ORCHESTRA Project for SARS-CoV-2"

_vaccines, 2023, doi:10.3390/vaccines11081361_

Round 1

Reviewer 1 Report

How European research projects can support vaccination strategies: the case of the ORCHESTRA project for SARS-CoV-2

Review:

The aim of this publication is to describe the early results of the EU-funded project ORCHESTRA, that combine international SARS-CoV-2 cohorts with particular focus on vaccine strategies.

The manuscript provides an interesting and concise overview of important results obtained throughout the ORCHESTRA study period, and is worthy of publication. That the ORCHESTRA study is impressive and provides high-powered results is without a doubt, however, the manuscript would benefit from placing more focus on the topics of vaccines, rather than on mainly focusing on the general strengths of the ORCHESTRA study.

Below I include some more specific points.

Abstract:

-      The abstract gives a good overview of the aim of the study; however, it would help the reader if some specific results could be included here.

Introduction:

-      The title and the aim were to present data that support vaccination strategies. From the introduction this is not that apparent. It would be helpful to include text that refers to specific vaccination strategies and issues relating to vaccine hesitancy.

Materials and methods:

-     ORCHESTA is an impressive data source combining data from several clinical cohorts. However, the description is rather generic of the cohort. The description would benefit from including methodologies focusing on vaccination uptake, efficacy and disease outcome; and how the data was standardized across the cohorts.

Results:

-          Again, it is an impressive data infrastructure. However, it is not quite clear from the text where the authors refer to up to 2 million individuals in patient registries, which is not apparent from Figure 1 where numbers do not add up to 2 million. Some more description of the patient registries would improve understanding of the manuscript.

-          Otherwise, the results give a concise overview of the results obtained through ORCHESTRA.

Discussion:

-          The authors state that ORCHESTRA counteracts selection bias by looking at an entire population group. None of the included cohorts, to the best of my knowledge, include the entire population of a country. Rather different subgroups from different countries are included. It would be clearer to the reader if this sentence (line 209-210) was modified to reflect this.

-          It would be advisable to include some discussion on the topics of vaccines and place the ORCHESTA study in context with other studies.

Author Response

Dear Reviewer, below our point-by-point response to your comments:

The aim of this publication is to describe the early results of the EU-funded project ORCHESTRA, that combine international SARS-CoV-2 cohorts with particular focus on vaccine strategies. The manuscript provides an interesting and concise overview of important results obtained throughout the ORCHESTRA study period, and is worthy of publication. That the ORCHESTRA study is impressive and provides high-powered results is without a doubt, however, the manuscript would benefit from placing more focus on the topics of vaccines, rather than on mainly focusing on the general strengths of the ORCHESTRA study. Below I include some more specific points.

Abstract:

  • The abstract gives a good overview of the aim of the study; however, it would help the reader if some specific results could be included here.

ANSWER: We want to thank the reviewer for the thorough revision of our manuscript. As suggested, some results of ORCHESTRA have been included in the abstract (please see lines 41-44)

Introduction:

  • The title and the aim were to present data that support vaccination strategies. From the introduction this is not that apparent. It would be helpful to include text that refers to specific vaccination strategies and issues relating to vaccine hesitancy.

ANSWER: Introduction was expanded to underline the value of ORCHESTRA for vaccine strategies. As a note, vaccine hesitancy is not a primary focus of the consortium, but the results can impact, through proper vaccine strategies, on vaccine uptake (lines 55-59 and 71-74)

Materials and methods:

  • ORCHESTA is an impressive data source combining data from several clinical cohorts. However, the description is rather generic of the cohort. The description would benefit from including methodologies focusing on vaccination uptake, efficacy and disease outcome; and how the data was standardized across the cohorts.

ANSWER: The methodology section has been expanded to describe the main direction of study of the project and standardization (please see lines 80-97). A supplementary table was added with the list of cohorts (Table S1).

Results:

- Again, it is an impressive data infrastructure. However, it is not quite clear from the text where the authors refer to up to 2 million individuals in patient registries, which is not apparent from Figure1 where numbers do not add up to 2 million. Some more description of the patient registries would improve understanding of the manuscript.

- Otherwise, the results give a concise overview of the results obtained through ORCHESTRA.

ANSWER: The introduction of patient registries in the results was not previously described in the methods. Additional details were added to the methods (please see lines 90-91) and in Table S1. Figure 1 has been updated including additional retrospective cohorts.

Discussion:

The authors state that ORCHESTRA counteracts selection bias by looking at an entire population group. None of the included cohorts, to the best of my knowledge, include the entire population of a country. Rather different subgroups from different countries are included. It would be clearer to the reader if this sentence (line 209-210) was modified to reflect this.

ANSWER: Indeed, the comment is correct. The sentence was improved to better describe the value of the cohort studies (please see lines 254-255).

It would be advisable to include some discussion on the topics of vaccines and place the ORCHESTA study in context with other studies.

ANSWER: a paragraph was added to include evidence from other studies that had a study design comparable with the ORCHESTRA study (please see lines 268-277)

Reviewer 2 Report

Overall, very good manuscript, currently readers may have to make some efforts to read the facts, so author could have used some more graphs/chart/tables to better elucidate. However, I still think it is good material to be published in current form. 

Author Response

Dear Reviewer, below a point-by-point response to your comments.

Overall, very good manuscript, currently readers may have to make some efforts to read the facts, so author could have used some more graphs/chart/tables to better elucidate. However, I still think it is good material to be published in current form.

ANSWER: We thank the reviewer for the positive comment. We tried to keep the communication as brief as possible. A table has been added in the supplementary (Table S1: “List of ORCHESTRA cohorts”).

Reviewer 3 Report

The authors present an appropriate design to explore risk factors, public health interventions, and vaccination effectiveness.

The term Post Covid syndrome is not really correct because of the heterogeneity of complaints - I would recommend the term Post Covid condition instead. Neurocognitive symptoms are more common than neurosensorial - under which phenotype do you summarize cognitive complaints?

Author Response

Dear Reviewer, below a point-by-point response to your comments.

The authors present an appropriate design to explore risk factors, public health interventions, and vaccination effectiveness.

The term Post Covid syndrome is not really correct because of the heterogeneity of complaints - I would recommend the term Post Covid condition instead. Neurocognitive symptoms are more common than neurosensorial - under which phenotype do you summarize cognitive complaints?

ANSWER: Many definitions were used to describe the health issues after acute SARS-CoV-2 infection, and we initially chose the one we were most familiar with. We agree with the Reviewer suggestion, which is in line with CDC and WHO. The text has been changed accordingly.

Reviewer 4 Report

Thank you for inviting me to review this short communication. The manuscript is a brief explanation and findings from the ORCHESTRA project for SARS-CoV-2. Overall it reads well but I think there are a few problems with this communication that should be improved:

1. The introduction might benefit from the justification of why the authors provided this manuscript and why we need to read this paper.

2. The methods need more information on samples and countries that participated in the cohort.

3. The results need subtitles for each objective that the authors indicated.

Author Response

Dear Reviewer, below a point-by-point response to your comments.

Thank you for inviting me to review this short communication. The manuscript is a brief explanation and findings from the ORCHESTRA project for SARS-CoV-2. Overall it reads well but I think there are a few problems with this communication that should be improved:

  1. The introduction might benefit from the justification of why the authors provided this manuscript and why we need to read this paper.

ANSWER: thank you for your comments. The introduction was expanded to explain better the focus of this paper (in particular lines 71-74)

  1. The methods need more information on samples and countries that participated in the cohort.

ANSWER: The Methods section was expanded and the list of cohorts participating in ORCHESTRA in Table S1 was added.

  1. The results need subtitles for each objective that the authors indicated.

ANSWER: Headings were added in the Result section to improve readability.

Reviewer 5 Report

The article is very interesting. I suggest several minor changes.

Please answer the following issues.

Does the project include patients with hematopoietic progenitor transplantation or treatment with anti-JAK drugs (Tofacitinib, Baricitinib, Upadacitinib, Ruxolitinib) as a risk group?

. Figure 1, (Figure 1. Overview of the ORCHESTRA project. )although interesting, does not provide sufficient information. The authors should include tables with all the cohort studies included in the project. One table would be for the general population, for example. In the table each row would be a cohort study, the country, the number of people, the start date of the cohort, and provide a bibliographic reference, if it exists, and a link to the cohort. The same should be done with the cohort of healthy persons, the frail population, etc. In each cohort, the main details should be provided, for example, cohort of Parkinson's patients or cohort of patients with solid organ transplantation, etc.

 Authors should refer to the ethical aspects. Has the project been approved by one or several ethical committees?

It would be interesting to be included within the text of the article an article link to the project web page https://orchestra-cohort.eu/

 The article's written text should be listed in the list of countries participating in each country one of the sub-projects. It is true that in figure one, the flags are included.

Author Response

Dear Reviewer, below a point-by-point response to your comments.

Does the project include patients with hematopoietic progenitor transplantation or treatment with anti-JAK drugs (Tofacitinib, Baricitinib, Upadacitinib, Ruxolitinib) as a risk group?

ANSWER: The “Fragile population cohorts” includes patients with hematopoietic progenitor transplantation as part of hematological cancer patients. Two cohorts include rheumatological patients (see the added Supplementary Table 1), but there are no inclusion or exclusion criteria regarding specific treatment.

Figure 1, (Figure 1. Overview of the ORCHESTRA project.) although interesting, does not provide sufficient information. The authors should include tables with all the cohort studies included in the project. One table would be for the general population, for example. In the table each row would be a cohort study, the country, the number of people, the start date of the cohort, and provide a bibliographic reference, if it exists, and a link to the cohort. The same should be done with the cohort of healthy persons, the frail population, etc. In each cohort, the main details should be provided, for example, cohort of Parkinson's patients or cohort of patients with solid organ transplantation, etc.

ANSWER: We agree with the reviewer's suggestion. We tried to summarize the requested data in Supplementary Table 1. The list is subdivided by type of cohort and includes name, organization, country, size, and if it started with ORCHESTRA or not. Unfortunately, the project did not publish a dissemination page for each of the cohorts, and we can’t provide a link for the cohorts. The numbers in Figure 1 changed because in the first draft we considered the number of patients included in major publications, while in the second version we considered all the patients enrolled in the cohorts.

Authors should refer to the ethical aspects. Has the project been approved by one or several ethical committees?

ANSWER:  All protocols were approved by the local Ethics Committee of participating sites. The ORCHESTRA project also underwent an ethics review as part of the evaluation process by the European Commission. We added this information in the Methods. (please see line 118-132)

It would be interesting to be included within the text of the article an article link to the project web page https://orchestra-cohort.eu/

ANSWER: The link was initially present as citation number two, and now it is also included in the main text (line 67).

The article's written text should be listed in the list of countries participating in each country one of the sub-projects. It is true that in figure one, the flags are included.

ANSWER: We added a Supplementary table with all the participating cohorts (Table S1), which includes the country as column.